# Prokaryotic responses to a warm temperature anomaly in northeast subarctic Pacific waters

Sachia J. Traving[1,9], Colleen T. E. Kellogg[2], Tetjana Ross[3], Ryan McLaughlin[4], Brandon Kieft[1], Grace Y. Ho [1,10], Angelica Peña [3], Martin Krzywinski[5], Marie Robert[3] & Steven J. Hallam [1,4,6,7,8 ✉]

Recent studies on marine heat waves describe water temperature anomalies causing changes in food web structure, bloom dynamics, biodiversity loss, and increased plant and animal mortality. However, little information is available on how water temperature anomalies impact prokaryotes (bacteria and archaea) inhabiting ocean waters. This is a nontrivial omission given their integral roles in driving major biogeochemical fluxes that influence ocean productivity and the climate system. Here we present a time-resolved study on the impact of a large-scale warm water surface anomaly in the northeast subarctic Pacific Ocean, colloquially known as the Blob, on prokaryotic community compositions. Multivariate statistical analyses identified significant depth- and season-dependent trends that were accentuated during the Blob. Moreover, network and indicator analyses identified shifts in specific prokaryotic assemblages from typically particle-associated before the Blob to taxa considered free-living and chemoautotrophic during the Blob, with potential implications for primary production and organic carbon conversion and export.

[1] Department of Microbiology & Immunology, University of British Columbia, Vancouver, BC V6T 1Z1, Canada. [2] Hakai Institute, Heriot Bay, BC V0P 1H0, Canada. [3] Institute of Ocean Sciences, Fisheries and Ocean Canada, Sidney, BC, Canada. [4] Graduate Program in Bioinformatics, University of British Columbia, Vancouver, BC V6T 1Z4, Canada. [5] Genome Sciences Centre, BC Cancer Agency, Vancouver, BC V5Z 4S6, Canada. [6] Genome Science and Technology Program, University of British Columbia, 2329 West Mall, Vancouver, BC V6T 1Z4, Canada. [7] Life Sciences Institute, University of British Columbia, Vancouver, BC V6T 1Z3, Canada. [8] ECOSCOPE Training Program, University of British Columbia, Vancouver, BC V6T 1Z3, Canada. [9] Present address: HADAL and Nordcee, Department of Biology, University of Southern Denmark, Campusvej 55, 5230 Odense M, Denmark. [10] Present address: Max Planck Institute for Marine Microbiology, Celsiusstraße 1, 28359 Bremen, Germany. ✉email: shallam@mail.ubc.ca

Extreme weather events including catastrophic hurricanes, floods, wild fires, and droughts appear to be increasing as mean global temperature rises[1,2]. While many of these impacts can be visibly recognized in terrestrial ecosystems[3–5], much less is known about the effects of heat waves on marine ecosystem functions and services. Marine heat waves (MHWs) are defined as discrete bodies of anomalous warm water reaching temperatures above the 90th percentile for more than five consecutive days[6]. A combination of atmospheric and oceanographic forces instigate MHWs, and climate change is predicted to increase the frequency and severity of these anomalies[7]. Several expansive MHWs have occurred in recent years[8,9]. One of the largest on record, colloquially known as the "Blob", was observed in the northeast subarctic Pacific Ocean (NESAP) manifesting sea surface temperatures 1–4 °C higher than average in 2014–2015[10–12] (Fig. 1a). Initially emerging in the upper 100 m of the water column, the temperature anomaly extended down to 200 m by 2015 before dissipating (Fig. 1b, c). Emergence and expansion of the Blob caused changes in food web structure including zooplankton and fish distribution patterns, and resulted in decreased chlorophyll a (Chl a), a proxy for primary production[13–15] (Fig. 1d).

Two domains of life underrepresented in MHW studies are bacteria and archaea. With their immense metabolic diversity, ubiquity, and abundance, these prokaryotic microorganisms are the major drivers of nutrient and energy flow within the microbial food web, also referred to as the microbial loop[16]. Changes in food web structure caused by MHWs could have important implications for marine biogeochemical cycles by altering the relationship between particulate and dissolved organic matter (OM) with concomitant impact on carbon (C) export processes[17]. Generally, prokaryotic communities are assumed functionally stable to incremental or episodic changes in their environments due to high diversity and large effective population sizes. However, recent experimental studies indicate potential sensitivity to ephemeral, warm water temperature anomalies. Joint and Smale used [3]H-leucine incorporation as a proxy for heterotrophic productivity across temperature gradients in the western English Channel and showed that episodically high temperatures can change nutrient and energy flow patterns through the microbial loop[18]. Thermal stress can also influence autotrophic microbes, and has been implicated in reduced productivity and photosynthetic efficiency in the dimethylsulfoniopropionate (DMSP)

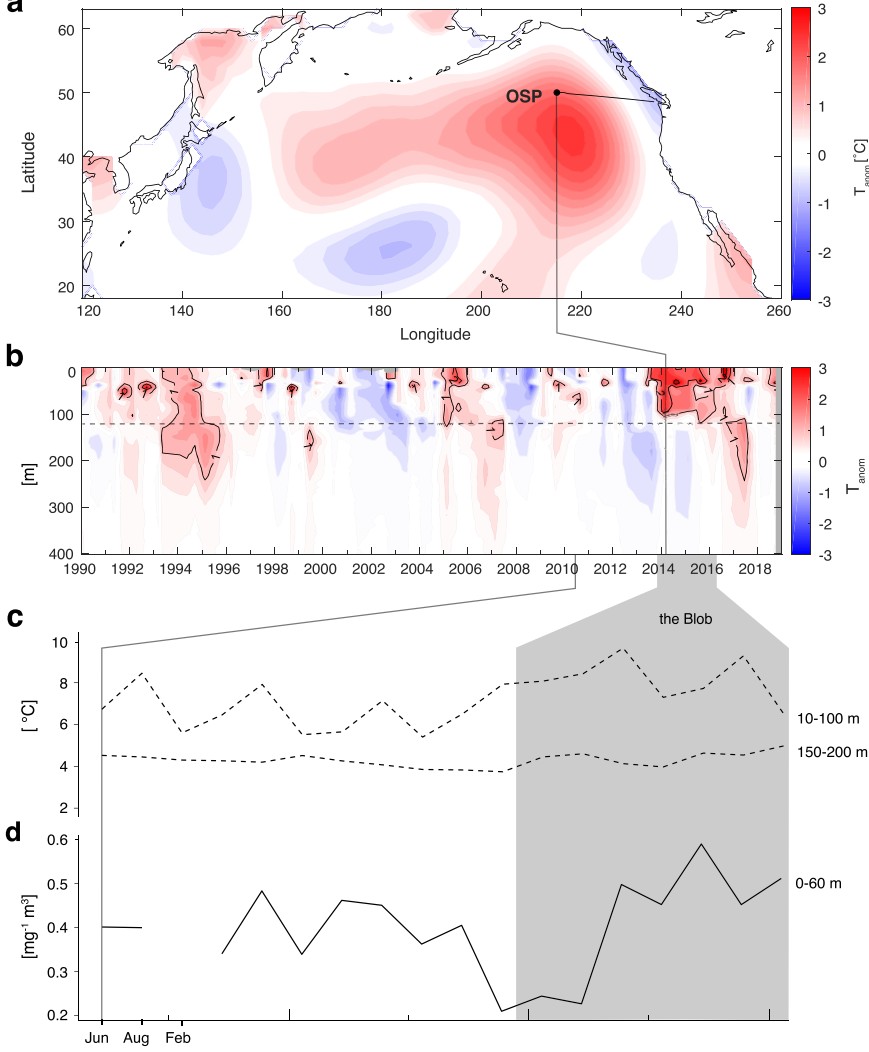

**Fig. 1 Geographic and physical description of the "Blob" marine heatwave. a** The geographic location of Ocean Station Papa (OSP, 50°N, 145°W) and the spread of the Blob in the NESAP on February 2014, shown as a temperature anomaly (°C) to the average mean (1981–2010) of P26 February climatology. **b** shows depth profiles of temperature anomalies (T, °C) for the upper 400 m of the water column at OSP, with the dotted line indicating the maximum depth of the mixed layer in the period shown. **c** shows the mean temperature for the upper 10–100 m and 150–200 m and the duration of the Blob highlighted in gray, over the period of this study; **d** is the average Chl a concentration in the upper 0–60 m of the water column.

producing dinoflagellate *Alexandrium minutum* in mesocosm experiments[19]. Interestingly, temperature induced changes in *A. minutum* activity corresponded to compositional changes in associated bacteria, reinforcing the potential impacts of MHWs on metabolic interactions[19].

Here we present a time-resolved study of prokaryotic community responses to the Blob in NESAP waters using small subunit ribosomal RNA (SSU or 16S rRNA) gene amplicon sequencing in combination with multivariate statistical methods. By comparing these amplicons with previously published geochemical[11] and pigment[13,14] data (also available at https://www.waterproperties.ca/LineP and https://open.canada.ca/data/en/dataset/871b0b32-3135-40c8-868e-c5d87800ca76), over a six-year time period before and during the Blob, we establish a baseline for prokaryotic communities at OSP. We go on to identify bacterial and archaeal indicators responding to Blob conditions and construct a conceptual model describing organic C conversion and export under MHW conditions based on contemporary understanding of the microbial loop in NESAP waters.

## Results and discussion

**Prokaryotic community structure at Ocean Station Papa**. Descriptions of prokaryotic community composition in the NESAP are based primarily on studies of bacterial and archaeal populations inhabiting specific depth intervals[20–23] and estimates of bulk production rates[24] with limited insight into temporal dynamics. In this study, we investigated changes in prokaryotic community structure at Ocean Station Papa (OSP or P26, 50°N, 145°W), the terminal station along the 1425 km Line P transect, to differentiate between seasonality and community responses to a marine heatwave. The Line P transect conducted by Fisheries and Oceans Canada is one of the oldest oceanographic time series in the world, traversing coastal to open ocean water conditions in the NESAP three times a year (typically in February, June, and August)[25]. Genomic DNA was extracted from 271 samples collected at OSP between June 2010 to February 2016, spanning 16 consistent depth intervals from surface to bottom waters. Sampled depths traversed discrete biogeochemical zones within the OSP water column (Fig. 2a, Fig. S1) which features a prominent oxygen minimum zone (OMZ) between 500–1500 m (Fig. S1e, f). The V4–V5 region of the SSU rRNA gene was amplified and sequenced resulting in a total of 41,951,488 quality-controlled reads clustered into 1310 operational taxonomic units (OTUs) at 97% sequence similarity used in downstream analyses (see "Methods").

As expected, prokaryotic communities clustered with depth closely following changes in biogeochemical zonation (Fig. 2a, b, Fig S2). These patterns were similar to those observed at Station ALOHA, the central sampling location of the Hawaii Ocean Time-series (HOT) in the North Pacific Subtropical Gyre[26]. Major taxonomic groups dominating OSP surface waters included Alphaproteobacteria, in particular the SAR11 clade, Cyanobacteria, Gammaproteobacteria, Bacteroidetes, and Actinobacteria (Figs. 3, 4), which correspond to broad level distribution patterns typically observed in surface ocean waters[27,28]. The mixed layer at OSP extends down to just below 100 m in winter months[29] and is shallower in summer. Accordingly, seasonal changes in community composition were observed in the surface mixed layer with winter communities between 10 and 100 m resolving a cluster separate from summer communities between 10 and 50 m (Fig. 3). Major phyla with strong seasonal shifts in relative abundance included

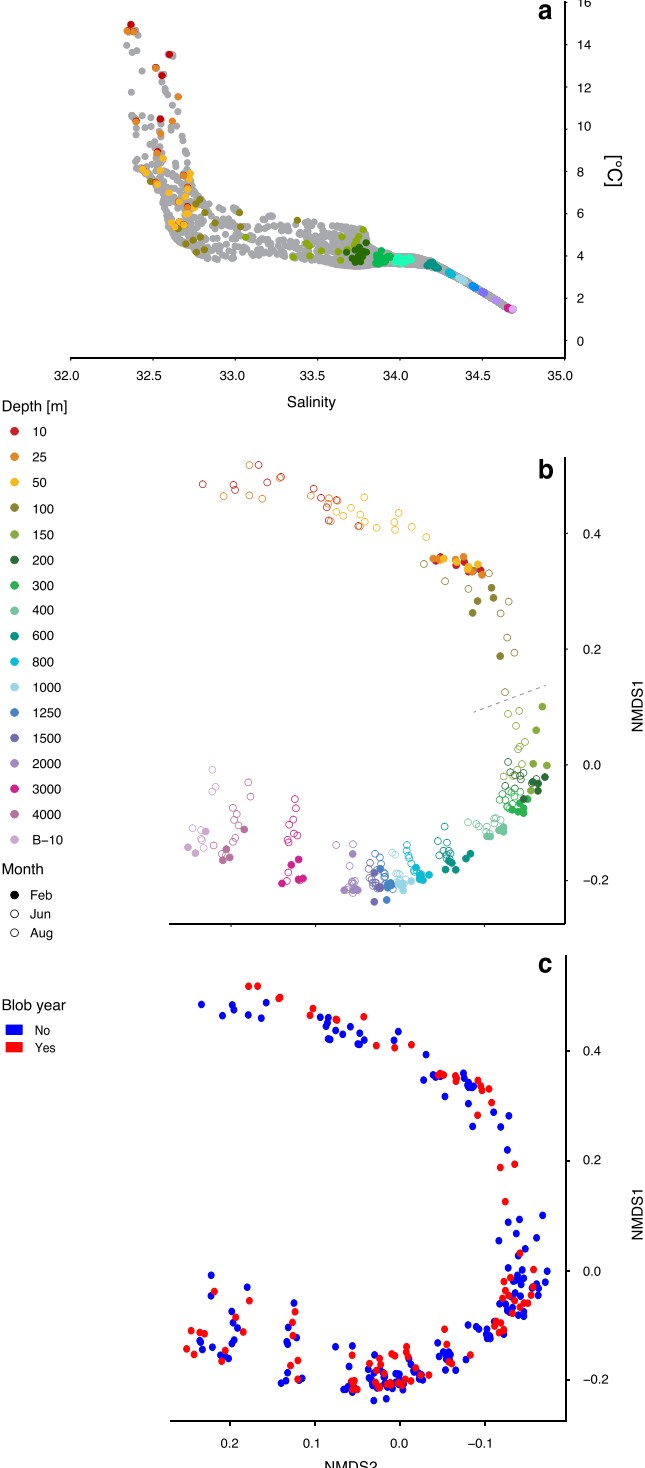

**Fig. 2 Microbial community composition and distribution in the NESAP water column. a** Temperature versus salinity plot (T-S) from full depth profiles representing the sampled hydrospace in the present study, with colored circles indicating microbial sampling depths. Bray-Curtis dissimilarity of all samples in a non-metric multidimensional scaling (nMDS) plot colored by **b** sampling depth and **c** whether it was a pre-Blob (No) or Blob (Yes) year. The dotted line on **b** mark the maximum mixed layer depth at Ocean Station Papa, as indicated on Fig. 1b.

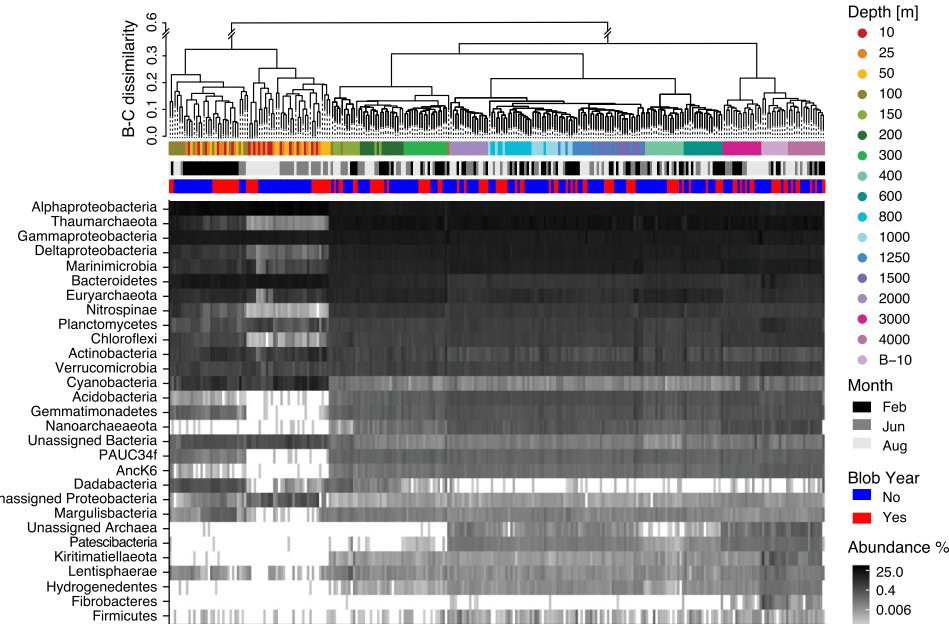

**Fig. 3 Prokaryotic community composition across the OSP microbial time series.** Dendrogram showing a hierarchical clustering analysis of all samples based on Bray-Curtis (B-C) dissimilarity, and suggest four major clusters (kmeans = 4, indicated by dotted line) with most depths forming separate subclusters of samples. First color-bar is sampling depth, middle bar is sampling month distinguishing between summer and winter months, and the last bar is whether the sample is from a pre-Blob (No) or Blob (Yes) year. The heatmap shows log transformed relative abundance of all phyla.

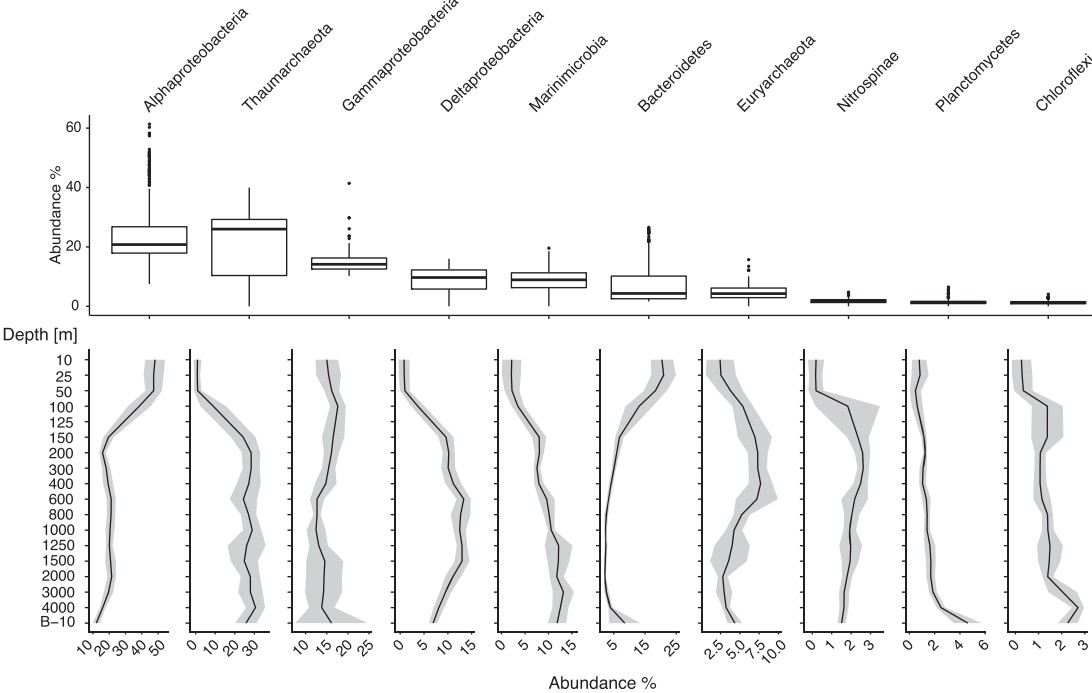

**Fig. 4 Depth distribution of the most abundant prokaryotic taxa.** The 10 most abundant phyla, with Proteobacteria resolved at Class level, ranked by total relative abundance across the entire dataset, shown as boxplots (centre line represents the median, the lower box limit the 25% quartile, the upper box limit the 75% quartile, and the whiskers extend to 1.5 * IQR). Relative abundance distribution along the vertical depth profile is shown below for each phyla. Depth profiles of phyla abundances partitioned after season and Blob/pre-Blob can be found in Fig. S8. The remaining phyla found in the present study is shown in Fig. S7, S9, and S10.

Thaumarchaeota, Deltaproteobacteria, Marinimicrobia, Nitrospinae, Chloroflexi, and Gemmatimonadetes which all increased in winter months (Fig. 3). This seasonal pattern is driven primarily by vertical mixing with deeper waters in winter months and increasing stratification, light levels, and phytoplankton production in summer months. Import of deep-water taxa into surface

waters is reflected in episodic increases in alpha diversity down to 100 m in winter months (Fig. S4). Interestingly, surface samples formed subgroups associated with pre-Blob and Blob conditions during both winter and summer months (Figs. 2c, 3) with concomitant changes in alpha diversity (winter alpha diversity was reduced during Blob conditions relative to pre-Blob

conditions, Fig. S4). Below 100 m alpha diversity appeared more stable across seasons while community composition shifted towards increased prevalence of chemoautotrophic taxa in line with changing biogeochemical gradients. Thaumarchaeota, particularly ammonia-oxidizing *Nitrosopumiales*, and Marinimicrobia[30,31] increased in relative abundance with depth, and Euryarchaeota and Deltaproteobacteria (particularly the SAR324 clade) peaked just above or within the OMZ (Fig. 4, Fig. S7–10). These observations are similar to those observed further north in the Gulf of Alaska OMZ[32] and to the south in the Eastern Tropical North Pacific (ETNP) OMZ[33].

Simply describing phylum-level changes in microbial composition fails to capture the full picture of microbial taxonomic stratification within the water column. This is especially evident when looking at the Proteobacteria, and specifically the Alpha- and Gammaproteobacteria classes. While some orders of Alphaproteobacteria decreased with depth (Rhodobacterales and Rhodospirallales, for example), SAR11 had both a surface peak in relative abundance, consisting primarily of Clade Ia, and as well as a subsurface peak between 200 and 600 m driven primarily by Clade II and to a lesser extent, Clade Ib. Similar depth stratification of SAR11 clades was observed in Monterey Bay[34]. A subsurface peak in SAR11 abundance was previously described at OSP[35], suggestive of metabolic capabilities suited for low oxygen environments. This has since been validated through the discovery of functional nitrate reductase (*nar*) genes in SAR11 from the ETNP OMZ[33,36], implicating SAR11 as potential driver of nitrogen loss in OMZs. Gammaproteobacteria consistently comprised ~10–15% of the community throughout the water column (Fig. 4), but the dominant orders shifted with depth. For example, SAR86, SAR92, and the OM43 clades were abundant in surface waters, sulfur-oxidizing Thiomicrospirales were abundant in the mesopelagic (peaking between 400–600 m), and orders such as Alteromonadales, Xanthomonadales, Pseudomonadales, Oceanospirallales were more abundant in deeper waters. Communities below 2000 m manifested higher relative abundances of Fibrobacteres, Nanoarchaeota, Patescibacteria, and unassigned Archaeal phyla (Fig. S7).

**Blob effects on phytoplankton.** Thermal conditions during the Blob resulted in enhanced water column stratification at OSP which reduced both vertical mixing and surface water nutrient concentrations[13]. Total Chl *a* declined between August 2013 and June 2014, the year immediately following Blob onset, and recovered to pre-Blob concentrations by August 2014[14]. At the same time, phytoplankton community composition shifted towards smaller cells in spring, with picophytoplankton such as Cyanobacteria and Chlorophytes making up greater proportions of the June 2014 and June 2015 biomass, respectively, compared to pre-Blob conditions[14]. Globally, oceanic cyanobacteria primarily consist of *Prochlorococcus* and *Synechococcus* populations, which thrive under low nutrient conditions and exhibit temperature and light driven niche-partitioning[37–39]. *Synechococcus* is the dominant cyanobacterium present in OSP waters with *Prochlorococcus* commonly unobserved (Fig. S8). However, *Prochlorococcus* were found in the amplicon data in February 2014 following the onset of the Blob, albeit at low levels (relative abundance 0.06–0.08% in the surface layer, Fig. S8). This hints that marine heat waves may favor the northward expansion of *Prochlorococcus*. Chlorophytes are a core constituent of nano- and picoplankton and contribute significantly to marine primary production[40]. Conversely, larger-celled Diatom and Pelagophyte populations declined between August 2013 and June 2015[13,14]. According to current models of marine primary production, changes in size distribution towards smaller phytoplankton could

result in decreased export of organic C from sunlit to dark ocean waters[41–43].

Many prokaryotes found in marine surface waters interact closely with phytoplankton populations provisioning their growth through the production of vitamins and other essential nutrients. Given the observed changes in phytoplankton composition during the Blob, we explored whether corresponding changes in bacterial and archaeal assemblages were observed, using an asymmetric indicator analysis[44,45]. We focused on two depth intervals spanning 10–100 m and 150–200 m (Fig. 1b) based on the emergence and subsequent deepening of the Blob over time at OSP as described by Freeland and Ross[11]. A total of 183 and 189 prokaryotic indicators were identified in the 10–100 m and 150–200 m zones, respectively. Additionally, co-occurrence networks were constructed for the communities in both of these depth intervals to estimate connectivity among prokaryotic taxa before and during the Blob. Following this, significant indicator OTUs (indicator value >0.7 and p-value < 0.05) were filtered based on their connectivity in the network (>3 positive edges), rather than a minimum relative abundance threshold. This approach allows for resolving potential key indicator OTUs without excluding conditionally rare taxa[46], and resulted in 134 indicator OTUs associated with pre-Blob and Blob conditions (Fig. 5, Fig. S6, and Table S1, S2) selected for further analyses and discussion.

**Prokaryotic indicators before the Blob.** Highly-connected indicator OTUs associated with pre-Blob conditions included a diverse array of Bacteroidetes, Alphaproteobacteria, Deltaproteobacteria, Gemmatimonadetes, Planctomycetes, and Nitrospinae (Fig. 5). Many lineages within these phyla, discussed in more detail below, are known to directly interact with growing phytoplankton or sinking particulate organic matter and have been implicated in cofactor biosynthesis (e.g., B-complex vitamins) or conversion of nitrogen (N) and sulfur (S) containing compounds produced during phytoplankton blooms. In addition to more persistent indicator OTUs, phytoplankton-associated dissolved organic matter (DOM) degraders from conditionally-rare taxa, such as the candidate phylum PAUC34f[47], were exclusive pre-Blob indicators.

Flavobacteriales within the Bacteroidetes are abundant marine particle- and bloom-associated bacteria that play a role in degrading high-molecular weight DOM[48]. Six pre-Blob Flavobacteriales indicators were affiliated with uncultivated NS2b and NS9 clades[49] (Fig. 5). In the North Sea, NS9 interact with flagellates and may subsist on cellular detritus or exudates[50]. Consistent with this observation, dinoflagellate populations were abundant at OSP under pre-Blob conditions but declined during the Blob[14]. Pre-blob indicators affiliated with Rhodobacterales (primarily *Rhodobacteraceae*[51,52]) within the Alphaproteobacteria were also identified. Rhodobacterales are known to associate with phytoplankton cells during blooms[53–55] and it has been posited that Rhodobacterales are primary $B_{12}$ producers for the phytoplankton community based on analysis of intact biosynthetic pathways harbored in isolate genomes[56]. The decline of these indicator OTUs may indicate uncoupling of essential metabolic interactions during the Blob.

In addition to organic C scavengers, several indicator OTUs affiliated with taxa known to play roles in cycling dissolved organic S compounds released by phytoplankton were identified, including Rhodospirallales and SAR11 subclade Ia within the Alphaproteobacteria[57] (Fig. 5). A single SAR11 subclade Ia pre-Blob indicator (OTU4) was the second most abundant SAR11 OTU identified within NESAP waters, making up more than 14.5% of the bacterial community between 10 and 100 m. While

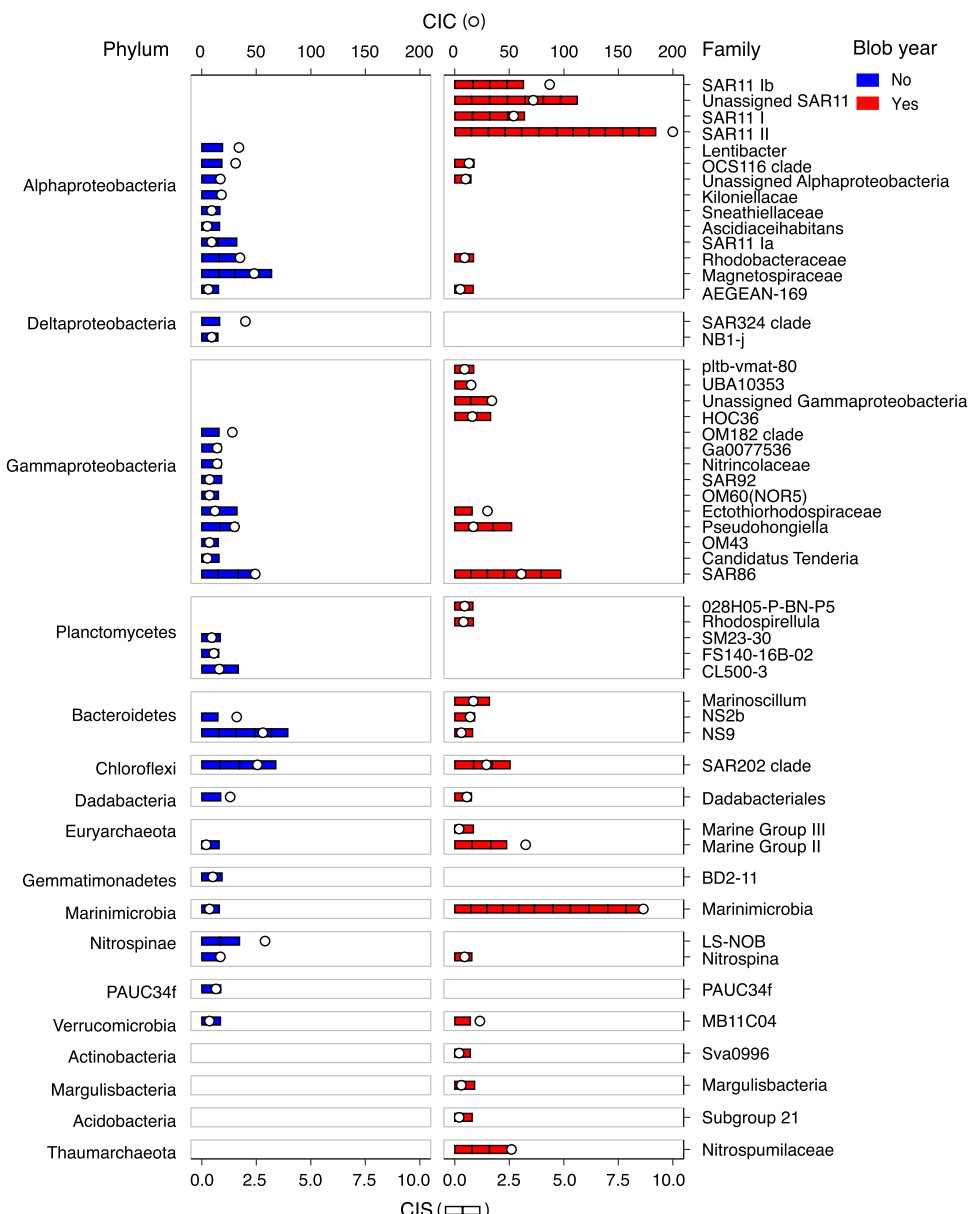

**Fig. 5 Taxonomic breakdown of indicator taxa.** Indicator OTUs for the Blob (right; red) and non-Blob (left; blue) years grouped into family-level taxa and quantified by cumulative indicator connectivity (CIC) and cumulative indicator score (CIS). CIC values (open points; top axis) represent the total weight of all network edges connected to indicator OTU nodes within the family-level taxonomic groupings on the right axis. CIS values (stacked bars; bottom axis) represent the total indicator score (sensitivity and specificity) of OTUs within a given family-level taxonomic grouping on the right axis, with each individual bar depicting the score of a single OTU.

more granular assignment of SAR11 clade Ia indicators based on 16S rRNA gene amplicon sequences alone is not possible, these OTUs could belong to the more cold-water ecotype (Ia.1), which may not fare as well during stratified heatwave conditions. In addition to Alphaproteobacteria, several less abundant pre-Blob indicators associated with organic S cycling including SAR324 and NB1-j from Deltaproteobacteria, and Gemmatimo-nadetes were also identified[58,59] (Fig. 5, Table S1, S2).

Several pre-Blob indicators, belonging to the Planctomycetes and Nitrospinae phyla were also identified. The Planctomycetes phylum appears to form three deep branches consisting of anammox Planctomycetes, Planctomycetacia, and Phycisphaera[60]. Planctomy-cetes indicators associated with pre-Blob conditions (Fig. 5) all appear to be related to the Phycisphaera which contain two marine isolates both isolated from alga and capable of nitrate to nitrite reduction[60,61]. On the other hand, one of the Planctomycetes Blob

indicators is related to Rhodopirellula (Fig. 5), a group typified by *R. baltica*. This species is capable of growing on N-acetylglucosamine, a key component of bacterial cell walls, as their sole C and N source[62] and its presence as a Blob indicator could suggest a potentially reduced reliance on phytoplankton-derived C within this phylum during marine heatwaves. Nitrospinae are autotrophic nitrite-oxidizers performing the second step of nitrification and appear to dominate this functional role in oxygenated ocean waters[63]. Unlike archaeal nitrifiers, Nitrospinae can utilize organic N in addition to ammonium to cover their N demand, which may in part explain their importance during pre-Blob conditions. Their decline during the Blob (or prevalence before the Blob) suggests that, as a consequence, nitrification may have been impacted during the Blob. Taken together with the observed shifts in phytoplankton community composition during the Blob, the decline of phyto-plankton- and particulate-associated indicator OTUs is consistent

with climate change induced changes in food web structure observed in other systems[64].

**Prokaryotic indicators during the Blob**. Highly-connected indicator OTUs that positively correlated with the Blob included OTUs affiliated with SAR11, Marinimicrobia, Nitrosopumilales, and Thermoplasmata (Fig. 5). These taxa are considered less dependent on direct phytoplankton interactions and particle association.

SAR11 had 29 highly connected indicator OTUs, 27 of which were correlated with the Blob. Compared to pre-Blob conditions, which was dominated by SAR11 subclade Ia, indicators belonging to SAR11 subclades Ib and II were overrepresented during the Blob (Fig. 5). Previous studies indicate that the biogeographic distribution of SAR11 subclades is correlated with latitudinal gradients with several subclades within the Ib group residing in warmer oligotrophic waters[65,66]. Current genomic evidence suggests that while the SAR11 clade is a functionally cohesive group[67] capable of assimilating large amounts of low molecular dissolved organic compounds, the distribution of genes encoding C assimilation pathways can vary considerably between subclades[66,68]. Therefore, changes in SAR11 distribution patterns have the potential to influence C conversion processes in surface waters.

Marinimicrobia and Nitrosopumilales had 15 highly connected indicator OTUs, 14 of which correlated with the Blob. Marinimicrobia contain multiple clades that partition along energetic gradients in the ocean with increased abundance in OMZs[20,21]. Several of these Marinimicrobia clades have been linked to sulfur oxidation and nitrous oxide ($N_2O$) reduction filling an important N cycle niche[20,23]. Furthermore, a recent study by Getz and colleagues[69] indicates that surface dwelling heterotrophic Marinimicrobia adapted to sunlight and oxygen manifest genomic streamlining, lower N content proteins and phototrophy. All six Marinimicrobia indicators found in the 10–100 m samples and five out of six indicators in the 150–200 m samples correlated with Blob (Table S1 and S2). Although our results do not resolve these indicators below the phylum level, sampling depth provides a basis for inferring traits as described by Getz and colleagues[69] consistent with aerobic heterotrophic metabolism.

Similar to Marinimicrobia, marine Thaumarchaeota contain both heterotrophic and chemoautotrophic members. Recent work highlights the presence of heterotrophic marine thaumarchaeota (HMT) in open ocean waters, with peak abundances around 200 m depth[70] at Station ALOHA in the North Pacific Subtropical Gyre. However, no HMT indicators were observed at OSP. The three Thaumarchaeota indicators observed in NESAP waters all correlated with the Blob and were affiliated with the ammonia oxidizing archaeal lineage *Nitrosopumilaceae*. Aerobic marine ammonia-oxidizing archaea (AOA) are key drivers of marine nitrification[71,72] and previous studies have observed abundance changes along temperature and nutrient gradients[73,74]. These observations suggest that the Blob favored AOA but did not significantly impact HMT. The increased abundances of both Marinimicrobia, which reduce $N_2O$, and *Nitrosopumilaceae* OTUs, which produce $N_2O$, during the Blob is especially interesting given recent work showing gene expression patterns suggestive of a metabolic coupling between members of these taxa[20].

Four out of five indicators from the Marine Group II and III (Themoplasmata) archaea were associated with the Blob (Fig. 5). These taxa play an important role in organic matter (OM) turnover, especially of phytoplankton-derived high molecular weight DOM. Both MGII and the less abundant MGIII contain surface-dwelling members adapted to a photoheterotrophic lifestyle[75,76]. MGII partition into free versus particle-attached groups[77], with free-living groups harboring traits for protein degradation[77] and photoheterotrophy[78]. MGII groups also partition seasonally, with MGII.B more abundant in the winter and MGII.A in spring and summer months[79]. MGII have been associated with increased phytoplankton abundance with corresponding growth rate stimulation in the presence of the picophytoplankton, Micromonas[77]. Prasinophytes, like Micromonas, increased in abundance during the Blob, which, combined with warmer and more stratified waters, has potential to select for specific MGII indicator OTUs. Even with the decline of some taxa known to associate with particles and phytoplankton during the Blob, the increase of MGII could continue to support degradation of high molecular weight DOM.

Among conditionally rare taxa, indicator OTUs affiliated with Actinobacteria, Acidobacteria, and Margulisbacteria were exclusively correlated with the Blob (Fig. 5). These bacterial groups are thought to carry out the metabolism of low molecular weight DOM. For example, the Actinobacteria indicator (Sva0996) readily transforms and assimilates dissolved organic N compounds (e.g., proteolysis products[77]), and the Margulisbacteria indicator is probably a close heterotrophic relative of Cyanobacteria with the capacity to oxidize simple C compounds for energy[80]. Taken together with the observed shifts in phytoplankton community composition during the Blob, the increase of free-living indicator OTUs is consistent with potential changes in OM composition and particle association in sunlit waters.

**Mixed responses and niche-differentiation**. Several bacterial groups exhibited mixed responses to the Blob which could be attributed to niche differentiation among ecotypes, similar to that described for SAR11 above. In the 150–200 m layer, four SAR202 indicator OTUs were associated with pre-Blob conditions while three SAR202 indicator OTUs were associated with the Blob (Fig. 5), supporting previous observations of niche partitioning within this clade[81]. Unfortunately, none of the SAR202 indicators could be resolved below Order level. Typically considered an inhabitant of the dark ocean, SAR202 has been observed in surface ocean waters during winter months[82,83] (Fig. 3) and the majority of all OTUs assigned to Chloroflexi (63/71) in the present dataset were annotated as SAR202. Genomic evidence suggests that SAR202 are sulfur oxidizers[84] and specialize in degradation of more recalcitrant OM[39]. A pangenomic analysis revealed the presence of seven SAR202 subclades and some of the surface clades contain rhodopsins[81]. It is possible that rhodopsin-containing clades may grow better in a stratified water column with more sustained access to light, as would have been the case during the Blob. Furthermore, with the decline in new production in the year following the onset of the Blob[14], it is also possible that OM in the surface ocean became more heavily processed selecting for prokaryotic groups capable of transforming recalcitrant OM.

Gammaproteobacteria are a common component of marine bacterioplankton[85] and were represented by 30 highly connected indicator OTUs. Two orders, SAR86 and Oceanospirillales represented 50% of all Gammaproteobacterial indicators, including three SAR86 OTUs correlated with pre-Blob conditions and six SAR86 OTUs correlated with the Blob (Fig. 5). Recent pangenomic analyses suggest that SAR86 ecotype differentiation may be, in part, driven by diversification of C utilization mechanisms, especially glycoside hydrolases[86,87]. Thus, C source may influence the expansion or decline of specific SAR86 ecotypes at OSP during the Blob. Oceanospirillales is capable of dissolved inorganic C assimilation and sulfur oxidation and has previously been credited with a significant role in dark ocean

primary production[88,89]. Three indictor OTUs affiliated with Oceanospirillales were associated with pre-Blob conditions and the Blob respectively, reinforcing the impact of MHWs on interactions driving coupled biogeochemical cycling in sunlit waters.

**Food web structure and C export.** The NESAP is an upwelling gyre characterized by high nutrient and low chlorophyll (HNLC) due to iron limitation on phytoplankton growth[90]. There is a marked seasonality in nutrient and primary production, but not in chlorophyll concentration, since microzooplankton are integral grazers, maintaining top down control of smaller phytoplankton cells[90,91]. Interannual variability in sea surface temperatures and nutrient concentrations are governed by El Niño/La Niña and the Pacific Decadal Oscillation, and the North Pacific Gyre Oscillation[91–93]. Ocean warming trends over the past fifty years are resulting in fresher and more nutrient-depleted NESAP surface waters, with declining subsurface oxygen concentrations and shoaling of the mixed layer depth[29,94,95]. Oxygen minimum zone expansion in the context of temperature anomalies such as the Blob have the potential to exacerbate changes in food web structure with resulting feedback on coupled biogeochemical cycling in NESAP waters.

Based on metabolic traits inferred from highly connected indicator OTUs across the time series, C conversion and export processes have potential to differ significantly between pre-Blob and Blob conditions. Using published values, publicly available data from the Line P Program, and the results of our amplicon sequencing of the prokaryotic community, we constructed a conceptual model of C flow in the OSP photic zone planktonic food web (Fig. 6, Supplementary Discussion). Where possible, we employed literature values to constrain C pools, but this was not always possible (Supplementary Discussion and Supplementary References). For example, the viral component was estimated based on published virus to prokaryote ratios because no values of viral abundance exist for this region that we are aware of. Furthermore, measurements of growth or production, grazing and C degradation very rare in the NESAP region and thus we did not attempt to constrain the magnitude of connections between pools, other than whether or not they would likely increase or decrease under marine heatwave conditions. These data gaps highlight areas for future research, some of which are currently being addressed by the NASA Export Processes in the Ocean from Remote Sensing (EXPORTS) program[96–98].

Although nanophytoplankton dominate phytoplankton biomass in NESAP waters, pigment fluxes indicate that microphytoplankton play a central role in C export in this region[85,99]. During the Blob, microphytoplankton abundance at OSP decreased and photosynthetically-derived C entered the food web primarily via prokaryotes and eukaryotic picophytoplankton, which have negligible sinking rates and likely resulted in decreased rates of C export to the deep ocean[14] (Fig. 6). This shift in phytoplankton communities to smaller cells is consistent with predictions for a future warmer ocean[17]. Indeed, changes in NESAP zooplankton were observed during the Blob, with a community shift towards smaller typically warm water taxa[100,101] (and negative anomalies in cold water taxa). With smaller zooplankton dominating the community during the Blob, both the composition and export of organic C was likely altered since smaller zooplankton produce smaller fecal pellets which are typically recycled within the photic zone[102] (Fig. 6).

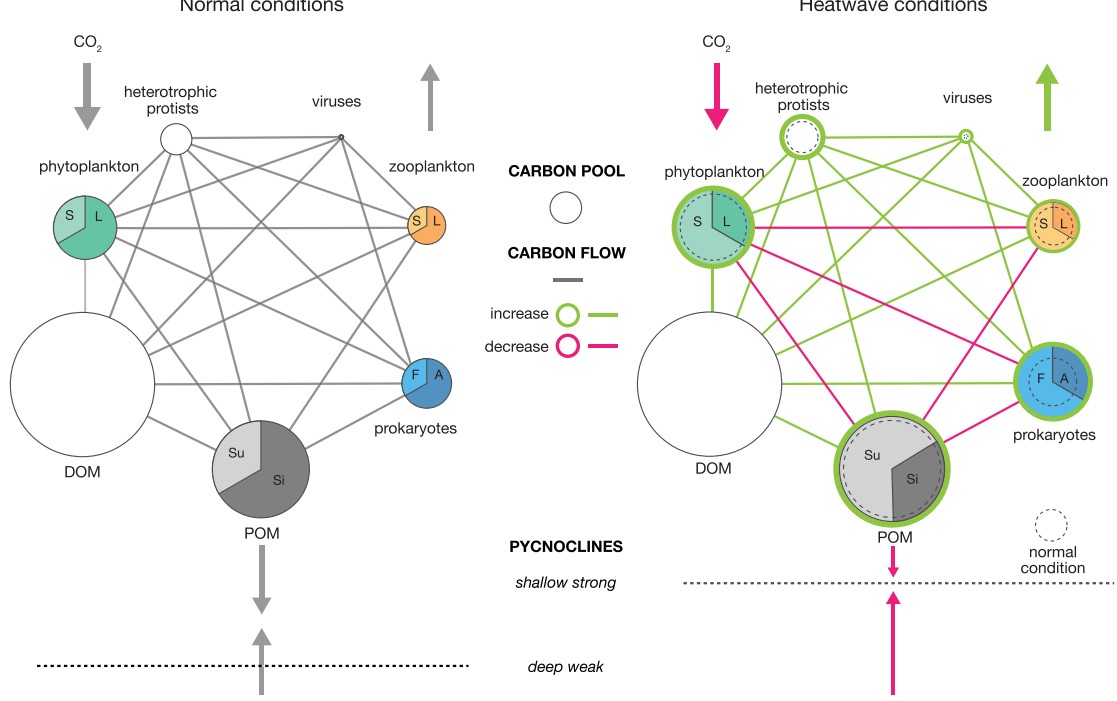

**Fig. 6 Conceptual model of NESAP food web structure and C export.** The food web during "normal conditions" (pre-Blob), here defined as periods with no MHW—on the left side; and changes associated with heatwave conditions—right side. The circles, or nodes, represent organic C or biomass pools. On the left side, nodes are scaled according to literature values for the different pools, see model description in Supplementary Material. For several nodes, size distribution (or niche in the case of prokaryotes) is displayed using colored pie charts, where S indicates "small", L indicates "large", A indicates "particle-associated", F indicates "free-living", Su indicates "suspended", and Si indicates "sinking". Shifts in pool sizes or composition are visualized using color proportions. Differences in pool sizes and C flow under normal and MHW conditions are indicated as a change in node size or line color in the right side of the model.

Interestingly, this interpretation of open ocean time series data is consistent with previous indoor mesocosm experiments using Baltic Sea surface waters in which elevating incubation temperatures by 2–6 °C increased organic C flow through the microbial loop[103].

Warmer waters are correlated with an increase in bacterial respiration[17] which would further contribute to increased C conversion. During the first year of the Blob, spring and summer net community production (NCP) remained within the normal range of values for the NESAP (1.6 and 2.2 mol C m$^{-2}$)[104]. While we did not measure respiration, a notable drop in NCP in 2015 ($-0.3$ mol C m$^{-2}$ in the spring, 0.3 mol C m2 in the summer m$^{-2}$)[104] despite increased phytoplankton biomass (and presumably primary production) is consistent with increased respiratory consumption of organic C. Recent work from the NASA EXPORTS program indicated that C turnover rates were influenced by both OM composition and prokaryotic community structure in controlled incubation experiments conducted with OSP waters in the dark at depth-calibrated temperatures[98]. Increased utilization of labile OM was associated with overrepresentation of taxa including Flavobacteriales (NS2b) and Alphaproteobacteria (Rhodobacterales)[98], which were indicators for pre-Blob conditions in our dataset.

The current conceptual model for OSP C conversion and export processes in response to MHWs requires additional parameter information to inform numerical model development. In addition to size-fractionated measurements of productivity, growth, and respiration it is imperative to link these parameters to multi-omic (DNA, RNA, protein, and metabolite) information. In particular, gene expression and metabolite profiling are needed to define functional relationships between trophic levels and reveal patterns of resource partitioning driving C conversion at elevated temperatures[105]. Moreover, single-cell amplified genome (SAG) sequencing and metagenome assembly has the potential to link the specific indicator groups reported here to inferred metabolic interactions within the microbial food web[106,107], and help resolve the observed differential responses of prokaryotic phylotypes to marine heatwaves. Already, based on the observed seasonal patterns in OSP surface communities presented here we can infer that surface communities contain functionally less redundant populations, as suggested by Furhman et al.[108]. These particular populations and associated functions may therefore be increasingly disturbed as MHWs become more intense and frequent. Ultimately, while the NESAP region contributes less to C export than other more productive regions[96] it represents a region so vast[109] that changes in nutrient and energy flow patterns could have a significant impact on marine ecosystem functions and services relevant to climate change. Continued time series observations are critical to our collective understanding of these impacts.

## Methods

**Sample collection**. The presented dataset consists of a total of 271 biological samples collected at Ocean Station Papa (P26, 50 °N, 145 °W) from CCGS John P. Tully during Line P cruises between June 2010 and February 2016. Sampling was conducted using a CTD rosette equipped with 24 10 L Niskin bottles. Microbial biomass was collected from 2 L seawater onto a 0.22 μm sterivex filter (Millipore Sigma, Darmstadt, Germany), fixed in 1.8 mL Sucrose Lysis Buffer (LSB), flash frozen in liquid N and stored at −70 °C until extraction. For a detailed filtration protocol please see Walsh et al.[110]. Temperature, salinity, and oxygen were measured by CTD. In addition to this, samples were collected for nutrients (nitrate plus nitrite, phosphate, an silica) analyzed on an Astoria Autoanalyzer[111], oxygen measured using Winkler titration, and phytoplankton pigment concentrations, analyzed using high performance liquid chromatography[112,113].

**Environmental DNA extraction**. DNA extraction was performed as described by Wright et al.[114]. In brief, 12.5 mg lysozyme (Sigma-Aldrich, Darmstadt, Germany) and 0.2 μg RNAse A were added to Sterivex filters and incubated at 37 °C for 1 h with rotation, followed by addition of 100 μL Proteinase K (600 mU, VWR, Canada) and 100 μL 20% sodium dodecyl sulfate (SDS), before incubation at 55 °C for 1–2 h. Lysate was removed by extrusion and DNA was extracted using 3 ml of phenol:chloroform:isoamyl alcohol mixture (IAA) (25:24:1). The DNA-containing aqueous layer was further purified with 3 ml of Chloroform:IAA (24:1) before being transferred onto a 10K Amicon filter cartridge (Millipore Sigma, Darmstadt, Germany), washed three times with Tris-EDTA buffer (pH 8), and concentrated to a final volume of 300–500 μL in Tris-EDTA by centrifugation (3500 × g), and stored at −80 °C. DNA concentrations were quantified using Quant-iT$^{TM}$ Pico-Green$^{TM}$ dsDNA Kit (ThermoFisher Scientific, Canada), and sent to the DOE Joint Genome Institute (JGI, California, USA) and used to generate 16S rRNA gene amplicon libraries of the V4–V5 region using the primers 515F-Y(5′-GTGYCA GCMGCCGCGGTAA-3′) and 926R (5′-CCGYCAATTYMTTTRAGTTT-3′)[115], following the JGI iTag Sample Preparation SOP (https://jgi.doe.gov/user-programs/pmo-overview/protocols-sample-preparation-information/) described by Rivers[116], before sequenced on an Illumina MiSeq, 2 × 300. Raw reads were merged using Fast Last Length Adjustment of Short Reads (FLASH) v1.2.11[117] with minimum and maximum overlap of 20 and 300 bp, respectively, and maximum mismatch of 30%. Merged reads from different libraries were pooled into one dataset based on sample names using the extract_barcodes_fastq.py script in QIIME1[10,13,118]. OTUs were generated by using non-singleton, quality-filtered reads from all libraries and clustering them at 97% similarity using the UPARSE algorithm[119] in USEARCH 8 v8.1.1861 with inherent chimera check. Additional reference-based chimera check was applied using UCHIME[120] and the SILVA 132 database[121]. Taxonomic assignment of OTUs was done using *feature_classify* and BLAST + wrapped in QIIME2[122] v2018. Chloroplast and mitochondrial reads were removed, leaving a total of 41,951,488 reads which clustered into 1,310 OTUs at 97% similarity. Mean sample size was 154,802 reads with the smallest and largest sample size being 44,681 and 268,405, respectively.

**Statistics and reproducibility**. All data analyses, unless otherwise described, were carried out in R[123] and Rstudio[124]. For detailed documentation including R packages and version numbers, please refer to the supplementary source data including a Rmarkdown file (Blob_MS_figures_v32.rmd) and input files (Blob_-markdown_input.zip) which can be downloaded from https://github.com/hallamlab/BLOB. An indicator analysis was performed on two subsets of the sequence data. The upper epipelagic communities (10–100 m) and the lower epipelagic communities (150–200 m) divided into pre-Blob (June 2010 to August 2013) and the Blob (February 2014 to February 2016) samples, as specified in Supplementary Data 2, column "BlobClus". Indicator OTUs with an association >0.7 and a *p*-value < 0.05 were retained for further analyses. Seven indicators were removed as they appeared in both analyses with opposite associations (OTU 449, 499, 633, 735, 753, 1069, and 1192). Network analyses were performed on the same subsets of the sequence data used for the indicator analyses, using CoNet[125] version 1.1.1.beta application installed within Cytoscape[126] version 3.7.1. For further details on the CoNet analyses, please see source data files Blob_settings.config, Blobsubset_depths.cys and Blob_subset_depths.gephi, which can be downloaded from https://github.com/hallamlab/BLOB. A summary of how the amplicon data was processed is shown in a workflow in Fig. S12.

**Reporting summary**. Further information on research design is available in the Nature Research Reporting Summary linked to this article.

## Data availability
DNA sequences of the 16S rRNA gene amplicons are available from the NCBI Sequence Read Archive under the BioProject no. PRJNA640752, accession no. SRR12059679-SRR12059949 (Supplementary Data 1).

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

## Acknowledgements

We would like to thank the captain, crew, and scientists onboard the CCGS John P. Tully for their extraordinary efforts in the field over many years, and Tijana Glavina del Rio at the DOE Joint Genome Institute (JGI) for project management on sample submission and sequencing. We also thank the many undergraduate helpers in the Hallam lab and ocean-going technical support staff including Jade Shiller and Chris Payne for their support in sample collection and processing. This work was performed under the auspices of the Scientific Committee on Oceanographic Research (SCOR), the US Department of Energy (DOE) Joint Genome Institute, an Office of Science User Facility, supported by the Office of Science of the U.S. Department of Energy under Contract DE-AC02- 05CH11231, the G. Unger Vetlesen and Ambrose Monell Foundations, the Natural Sciences and Engineering Research Council of Canada, the Canada Foundation for Innovation, and the Canadian Institute for Advanced Research through grants awarded to S.J.H. S.J.T. was supported by the Danish Research Council (grant DFF-7027-00043B) and CTEK by the Tula Foundation.

## Author contributions

S.J.T., C.T.E.K. and S.J.H. conceived the study. T.R., A.P., M.R. and G.Y.H. collected and processed samples. S.J.T., C.T.E.K., B.K., R.M., G.Y.H., T.R., A.P., M.K. and S.J.H. analyzed the data and/or provided graphical interpretation of data. S.J.T., C.T.E.K., B.K. and S.J.H. wrote the manuscript. All authors have read and contributed to the text.

## Competing interests

S.J.H. is a co-founder of Koonkie Inc., a bioinformatics consulting company that designs and provides scalable algorithmic and data analytics solutions in the cloud. The remaining authors declare no competing interests.
