## [Transparent Peer Review File · Communications Biology]

Reviewers' comments:

Reviewer #1 (Remarks to the Author):

Review for Traving et al Comm Biol

The authors use SSU rRNA amplicon analysis to assess the impact of a marine heat wave that took place in the North Pacific in 2014-2015 (the blob). By comparing microbial community composition before, after, and during the blob the authors are able to assess the impact of this marine heatwave on overall community dynamics. A large set of samples are analyzed from multiple depths over a multi-year period. Statistical methods are used to assess changes in community structure.

This work fits with findings that climate change is leading to increased stratification of ocean surface waters and the expansion of OMZ (Polovina *Geophys Res Lett* 2008; Stramma et al *Science* 2008). The article is well-written, the methods are robust, and some important information on how community structure shifted during the blob is presented. Some may question the impact of this work because it is 16S based (rather than relying on community metagenomics), but amplicon libraries are appropriate for many of the questions posed here, and they remain an important tool in the toolbox of microbial ecologists. The large number of samples analyzed and the length of the time-series are real strengths of this manuscript. A nice conceptual overview is provided to put lots of different pieces together- some of the discussion on exports rates are speculative, but the authors describe the caveats here and note the need for future studies using other approaches. Overall this is well-done and an important contribution. I only have a few general suggestions that may help clarify a few things.

One caveat of this study is that the phylogenetic groups that are analyzed are quite broad and sometimes encompass lineages with disparate physiologies. This makes it a bit more difficult to interpret the results of the indicator analyses. The authors did a good job examining specific subclades of SAR11 and discussing this issue for SAR202- if a bit more of that could be done for other groups I think that could really help clarify some of the exciting patterns that are revealed here. For example, the Marinimicrobia are a phylum-level group- they encompass some groups with the sulfur and nitrous oxide metabolic activities mentioned, but there are also some streamlined epipelagic heterotrophs in this group (Getz et al *mBio* 2018). Similarly, there are some interesting patterns shown here for the Thaumarchaeota, and some OTUs for these groups are "indicator OTUs" during blob conditions- can the authors distinguish between the AOA OTUs and those that belong to the heterotrophic thaumarchaeota (sometimes called the pSL12 group, or UBA057 in the GTDB) ? It would be interesting to make this distinction since the two groups have very different metabolism (Aylward Santoro *msystems* 2020; Reji Francis *ismej* 2020). At ALOHA the heterotrophic thaumarchaeota appear to peak in abundance around 200m- it would be interesting to know if that is the case here too, and how they responded to the blob.

There is not that much discussion in the text about the prominent OMZ that is encompassed by this dataset, which I thought was a bit odd since increased stratification is also thought to lead to larger OMZs (Stramma et al *Science* 2008). It seems like this dataset would be a wonderful opportunity to examine how/if the OMZ shifted. In deeper waters did it look like the OMZ and associated taxa expanded during the blob? Or am I misunderstanding some reason why the OMZ wasn't discussed more?

Just a bit of a side note really - I was a bit surprised how similar the results are here for the results of Fig. 2 and 3 and those reported for metagenomic community structure at ALOHA (Fig 1 Mende et al *Nat Microbiol* 2017). The curved structure of the NMDS are the same, and the dendrograms are highly similar except for the difference in samples. This just reinforces the view that these patterns are quite clear and reproducible. It might be worth mentioning this when these patterns are first presented (line 121) just to show that this is a very well-described pattern that shows up when analyzing even very

different data types (16S vs metagenomic). It supports the view that the 16S datasets appear to be pretty good quality.

Small methodological point - It seems the authors used OTUs clustered at 97% identity for the analyses here- I don't think this would change the overall results much, but it's worth noting that many in the field are moving towards analysis of ASVs using methods like DADA2 (Callahan et al ismej 2017), which in theory give higher resolution since fine-scale differences in 16S variants can be detected. I don't think any new analyses need to be done, but the authors may wish to keep these in mind for future work if fine-scale variability is of interest.

Line 149; please italicize genus names

Line 268: missing words here- "gammaproteobacteria are a common component of marine bacterioplankton and"

Frank Aylward

Reviewer #2 (Remarks to the Author):

Traving and colleagues present an analysis of the prokaryotic community in the northeast subarctic Pacific Ocean with a focus on the time around the temperature anomaly known as the blob. Even though the data type is slightly outdated (16S rRNA), the data generated in this study has the potential to enable great insight into the microbial dynamics of a marine temperature anomaly. The authors find that depth- and season-dependent trends were accentuated during the temperature anomaly and that microbial populations tend to be considered free-living and chemoautotrophic during the Blob.

I find the authors' conclusions intriguing but due to the narrative style of the results, it remains unclear how exactly the conclusions are supported by the data and which data was generated in this study (vs available from previous studies). Further, the discussion of these results in comparison to other findings and the influence of marine temperature anomalies should be improved.

In general, I would appreciate the mention of more specific results and data in the text. This should further be supported by more focused figures.

Figures

Figure 1:

Fine in my opinion

Figure 2:

Part A is too small to be interpreted properly

Part B merely shows a gradient across depth, there are better ways to display this than an nMDS biplot. Further, there is an unreferenced marking.

Part C-F: Since these are merely recoloring of part B, these should have a distinct purpose. Yet, the most important part for this manuscript (D) shows that the "blob" is a minor factor in partitioning the whole sample set. (This is expected as depth is expected to be the main driver and that is shown in A, so a better visualization would be preferred. Maybe depth specific plots or so)

Figure 3:

This figure uses the information used for Fig. 2 and displays it as a heatmap + clustering. This figure is more information-dense and in my opinion, could replace figure 2 making room for a blob-specific figure of the microbial community.

Figure 4:

It is very confusing that down is more. Usually, similar plots signify higher abundances as vertical up. Further, I'm not sure what the purpose of the boxplots on the right side is. Why would you show average over the whole dataset after showing that season and depth have a large impact on community composition? This figure could better be used to display some of the results of the network analysis (in a tractable way)

Figure 5:

It seems ok but I would propose to either remove the redundancy as the network on the right already displays all changes or annotate both sides.

Specific comments:

Line 144:

"August 2013 and June 2014, the year immediately following Blob onset, and recovered to pre-Blob concentrations by August 2014."

Even though this is a citation, the comparison between June 2013 and 2014 would be more interesting to understand seasonality vs "blob" effects

Line 168:

This approach resolved 134 indicator OTUs associated with pre-Blob and Blob conditions (Fig. S4, S5, and Supplementary Table 1, 2) many of which would have been omitted solely on the basis of abundance filtering (Fig. S4).

This sentence needs additional explanation. Why would these OTUs be excluded? What are these indicator OTUs indicative for?

Line 175:

"Many of these taxa are known to directly interact with growing phytoplankton or sinking particulate organic matter and have been implicated in cofactor biosynthesis (e.g., B complex vitamins) or conversion of nitrogen (N) and sulfur (S) containing compounds produced during phytoplankton blooms."

The problem with reporting results on the phylum level is that many microbial species have the mentioned traits both within these phyla and outside and vice versa many members of these phyla have completely different traits. Would it be possible to pinpoint at least specific genera to support these interpretations? Currently, it seems that possibly unrelated examples from these phyla are discussed.

Line 209:

See line 175. This section suffers from the same problem as the one above.

Line 280:

The section "Food web structure and carbon export" would benefit strongly from an integration of the previously mentioned results. Which specific new results were used to design the conceptual model? Which parts of the model can be quantified? Which ones cannot right now?

ECOSCOPE
routing the microcosmos

THE UNIVERSITY OF BRITISH COLUMBIA

Referee expertise:

Referee #1: microbial diversity, genomics, marine bacteria, environmental microbiology

Referee #2: bacterial communities, metagenomics, ocean microbiome

Reviewers' comments:

Reviewer #1 (Remarks to the Author):

The authors use SSU rRNA amplicon analysis to assess the impact of a marine heat wave that took place in the North Pacific in 2014-2015 (the blob). By comparing microbial community composition before, after, and during the blob the authors are able to assess the impact of this marine heatwave on overall community dynamics. A large set of samples are analyzed from multiple depths over a multi-year period. Statistical methods are used to assess changes in community structure.

This work fits with findings that climate change is leading to increased stratification of ocean surface waters and the expansion of OMZ (Polovina Geophys Res Lett 2008; Stramma et al Science 2008). The article is well-written, the methods are robust, and some important information on how community structure shifted during the blob is presented. Some may question the impact of this work because it is 16S based (rather than relying on community metagenomics), but amplicon libraries are appropriate for many of the questions posed here, and they remain an important tool in the toolbox of microbial ecologists. The large number of samples analyzed and the length of the time-series are real strengths of this manuscript. A nice conceptual overview is provided to put lots of different pieces together- some of the discussion on exports rates are speculative, but the authors describe the caveats here and note the need for future studies using other approaches. Overall this is well-done and an important contribution. I only have a few general suggestions that may help clarify a few things.

One caveat of this study is that the phylogenetic groups that are analyzed are quite broad and sometimes encompass lineages with disparate physiologies. This makes it a bit more difficult to interpret the results of the indicator analyses. The authors did a good job examining specific subclades of SAR11 and discussing this issue for SAR202- if a bit more of that could be done for other groups I think that could really help clarify some of the exciting patterns that are revealed here. For example, the Marinimicrobia are a phylum-level group- they encompass some groups with the sulfur and nitrous oxide metabolic activities mentioned, but there are also some streamlined epipelagic heterotrophs in this group (Getz et al mBio 2018). Similarly, there are some interesting patterns shown here for the Thaumarchaeota, and some OTUs for these groups are "indicator OTUs" during blob conditions- can the authors distinguish between the AOA OTUs and those that belong to the heterotrophic

DR. STEVEN HALLAM • UNIVERSITY OF BRITISH COLUMBIA • ECOSCOPE PROGRAM • 2552-2350 HEALTH SCIENCES MALL,
VANCOUVER, BC V6T 1Z3 CANADA • (604) 827-3420 • FAX (604) 822-6041 •
E-MAIL shallam@mail.ubc.ca • WEBSITE <https://hallam.microbiology.ubc.ca>

thaumarchaeota (sometimes called the pSL12 group, or UBA057 in the GTDB) ? It would be interesting to make this distinction since the two groups have very different metabolism (Aylward Santoro msystems 2020; Reji Francis ismej 2020). At ALOHA the heterotrophic thaumarchaeota appear to peak in abundance around 200m- it would be interesting to know if that is the case here too, and how they responded to the blob.

Response: Thank you for the in-depth review of our paper. We appreciate your inquiry if the indicator Thaumarchaeota OTUs could possibly be HMT and not AOA. This is a great question that we had not investigated in depth. BLAST best hits from NCBI resulted in 92-93% percent identity to *N. cobalaminigenes* and *N. ureiphilus*, both of which are AOA). We then queried these against IMG/M and found they are 93-95% similar to Candidatus *Nitrosopelagicus brevis* CN25, another AOA. We then pulled the 16S rRNA genes from some of the MAGs described in Aylward and Santoro 2020 - specifically from AAIW, AABW, NADW (with the UBA57 lineage), an HMT 16S from ALOHA at 200 m (EF106858.1) and also included *N. brevis* CN25 – and queried the Thaumarchaeota indicator OTUs against these reference sequences. They were only 81% similar to HMT 16S genes (but again 93-95% similar to *N. brevis*). Based on this comparison, it seems the Blob had more of an impact on AOA than HMT, at least as far as our analysis shows. We have also worked on the text regarding the Thaumarchaeota identified in our indicator analyses.

Regarding Marinimicrobia, this is another great point and something we definitely thought about when drafting this manuscript. The Getz et al. reference was very interesting, thank you, and we've cited it in the text to add a bit more nuance to the discussion. Still, with the poor taxonomic resolution we obtained specifically on these Marinimicrobia indicators (Phylum-level) we have kept the discussion to moderate. We believe a phylogenomic approach would be necessary to unpack this phylum further and hope that this can be done in the future. If the reviewer deems it necessary we would be happy to provide consensus reads for all identified indicator OTUs in the supplementary material, so any future studies resolving these lesser known groups, could shed new light on our findings.

There is not that much discussion in the text about the prominent OMZ that is encompassed by this dataset, which I thought was a bit odd since increased stratification is also thought to lead to larger OMZs (Stramma et al Science 2008). It seems like this dataset would be a wonderful opportunity to examine how/if the OMZ shifted. In deeper waters did it look like the OMZ and associated taxa expanded during the blob? Or am I misunderstanding some reason why the OMZ wasn't discussed more?

Response: Thank you for bringing this up. During the Blob, there was no evidence of notable OMZ expansion above and beyond the current average annual rate of expansion, especially into the depth range that was directly impacted by The Blob (0 – 300 m; see temperature anomalies highlighted in Fig. 1b and also Freeland and Ross, 2019). While it is tempting to also discuss the OMZ community in detail here, we refrained, to keep the focus of this manuscript on changes in the microbial community

in waters known to be *directly* influenced by The Blob. The OMZ community is a great topic to tackle in a subsequent manuscript and with samples collected post 2016.

Just a bit of a side note really - I was a bit surprised how similar the results are here for the results of Fig. 2 and 3 and those reported for metagenomic community structure at ALOHA (Fig 1 Mende et al Nat Microbiol 2017). The curved structure of the NMDS are the same, and the dendrograms are highly similar except for the difference in samples. This just reinforces the view that these patterns are quite clear and reproducible. It might be worth mentioning this when these patterns are first presented (line 121) just to show that this is a very well-described pattern that shows up when analyzing even very different data types (16S vs metagenomic). It supports the view that the 16S datasets appear to be pretty good quality.

Response: thank you for pointing out that we missed this reference. We have cited the mentioned work by Mende et al. (2017) and made the following changes to the sentence, based on the reviewer's suggestion: "As expected, prokaryotic communities clustered with depth closely following changes in biogeochemical zonation (Fig. 2, S2), and exhibited broad patterns in community composition similar to what previously have been reported from station ALOHA²⁷."

Small methodological point - It seems the authors used OTUs clustered at 97% identity for the analyses here- I don't think this would change the overall results much, but it's worth noting that many in the field are moving towards analysis of ASVs using methods like DADA2 (Callahan et al ismej 2017), which in theory give higher resolution since fine-scale differences in 16S variants can be detected. I don't think any new analyses need to be done, but the authors may wish to keep these in mind for future work if fine-scale variability is of interest.

Response: we agree with the reviewer that ASVs currently seems to be the preferred method in the 16S amplicon field. However, we kept with the 97% clustering because i) as the reviewer points out themselves, we do not believe overall patterns or results discussed here will change significantly if we employed a denoising / ASV approach, and hence would not be worth re-analyzing the entire dataset from scratch, and ii) we would still need to be cautious to not over-interpret taxonomic groups at the ASV level at Ocean Station Papa, as it is a less studied region in terms of linking 16S amplicon identities with more thorough analyses of underlying fine scale phylogenetic diversity, ecotypes and functional potential.

Line 149; please italicize genus names

Response: corrected, thank you.

Line 268: missing words here- "gammaproteobacteria are a common component of marine bacterioplankton and"

Response: corrected, thank you.

Reviewer #2 (Remarks to the Author):

Traving and colleagues present an analysis of the prokaryotic community in the northeast subarctic Pacific Ocean with a focus on the time around the temperature anomaly known as the blob. Even though the data type is slightly outdated (16S rRNA), the data generated in this study has the potential to enable great insight into the microbial dynamics of a marine temperature anomaly. The authors find that depth- and season-dependent trends were accentuated during the temperature anomaly and that microbial populations tend to be considered free-living and chemoautotrophic during the Blob.

I find the authors' conclusions intriguing but due to the narrative style of the results, it remains unclear how exactly the conclusions are supported by the data and which data was generated in this study (vs available from previous studies). Further, the discussion of these results in comparison to other findings and the influence of marine temperature anomalies should be improved.

Response: Thank you for the in-depth review. We have gone through the text and indicated when we are presenting new versus published data from Ocean Station Papa. For example, we have clarified at the end of the introduction that the geochemical and pigment data has previously been published and referenced accordingly. It now reads: “Here we present a time-resolved study of prokaryotic community responses to the Blob in NESAP waters using small subunit ribosomal RNA (SSU or 16S rRNA) gene amplicon sequencing in combination with multivariate statistical methods. By comparing these amplicons with previously published geochemical^{11,20} and pigment^{13,14} data (also available at <https://www.waterproperties.ca/LineP> and <https://open.canada.ca/data/en/dataset/871b0b32-3135-40c8-868e-c5d87800ca76>) over a six-year time period before and during the Blob, we establish a baseline for prokaryotic communities at OSP. We go on to identify bacterial and archaeal indicators responding to Blob conditions and construct a conceptual model describing organic carbon conversion and export under MHW conditions based on contemporary understanding of the microbial loop in NESAP waters.” Another example is in the discussion, which now reads: “While we did not measure respiration concurrently with the collection of the amplicon data, Bif and Hansell¹⁰⁰ have reported on the net community production (NCP) in NESAP between 1988-2017, based on measurements and estimates using Argo float data. They report that in the first year of the Blob IN 2014, the NCP was estimated within normal range values for the NESAP (1.6 and 2.2 mol C m⁻²), followed by a notable drop on NCP in 2015 (-0.3 mol C m⁻² in the spring, 0.3 mol C m² in the summer m⁻²), despite an increased phytoplankton biomass (and presumably primary production) in this period, which is consistent with increased respiratory consumption of organic carbon.” Similarly, we have gone through the discussion to make sure to bring back in current literature available on marine heatwaves and microbial communities; which is extremely limited. As stated in the introduction, most published studies have focused on permanent but incremental temperature increases.

In general, I would appreciate the mention of more specific results and data in the text. This should further be supported by more focused figures.

Response: this was also highlighted by reviewer 1. We have gone through the text and added more detail in particular to the section on the Marinimicrobia, Planctomycetes, and Nitrosopumilaceae. We have also made a new graph, Figure 5 in the revised manuscript for a better presentation of the indicator OTU results. Together, we hope that this has improved the narrative flow with renewed emphasis on key results in figures as described below.

Figures

Figure 1: Fine in my opinion

Figure 2: Part A is too small to be interpreted properly. Part B merely shows a gradient across depth, there are better ways to display this than an nMDS biplot. Further, there is an unreferenced marking.

Response: thank you for highlighting the unassigned asterisk. We have explained its intention in the figure legend. It is intended to mark the maximum depth of the mixed layer (which we also show in figure 1).

Part C-F: Since these are merely recoloring of part B, these should have a distinct purpose. Yet, the most important part for this manuscript (D) shows that the "blob" is a minor factor in partitioning the whole sample set. (This is expected as depth is expected to be the main driver and that is shown in A, so a better visualization would be preferred. Maybe depth specific plots or so)

Response: we have removed what was the original panel C, E and F, and moved them to supplementary material, and increased the size of panel A (now "a"). We have however, kept panel B and D (now panel b and c) of the nMDS plot colored based on whether a sample was from a non-Blob or Blob year. This was done to accommodate comments from Reviewer 1 about results on whether any disturbances can be observed below surface waters, due to the Blob. We hope the edits still has helped the overall impression and utility of figure 2. Furthermore, Reviewer 1 was enthusiastic about the inclusion of this figure, given similarities to to patterns observed in Mende et al Nat Microbiol 2017. We therefore advocate for continued inclusion of this figure, albeit reduced in scope with your helpful comments. Regarding depth specific plots, these are provided in supplementary material, figure S1, and we have further added information on profile differences between Blob and non-Blob samples for each parameter.

Figure 3: This figure uses the information used for Fig. 2 and displays it as a heatmap + clustering. This figure is more information-dense and in my opinion, could replace figure 2 making room for a blob-specific figure of the microbial community.

Response: please see response to comments regarding Figure 2. In short, we have reduced Figure 2 based on your feedback, but retained parts if it as Reviewer 1 found merit in those parts of the figure, as it is a commonly used type of plot for microbial community composition and is easy to relate to for others working with similar type data. We are glad to see your approval of Figure 3, and also appreciate and will heed your call for a Blob-specific figure (as mentioned above).

Figure 4: It is very confusing that down is more. Usually, similar plots signify higher abundances as vertical up. Further, I'm not sure what the purpose of the boxplots on the right side is. Why would you show average over the whole dataset after showing that season and depth have a large impact on community composition? This figure could better be used to display some of the results of the network analysis (in a tractable way)

Response: we apologize that the figure might be misinterpreted due to being flipped 90 degrees compared to the legend text. We have turned the figure and also rephrased the figure legend to make it easier to understand what is being represented.

Figure 5: It seems ok but I would propose to either remove the redundancy as the network on the right already displays all changes or annotate both sides.

Response: This is now Figure 6. We have made minor alterations to the figure by annotating the right side. Thank you for this suggestion. We do understand the suggestion to reduce it to a single food-web, but find it harder to then convey their environmental differences, e.g. pycnocline, CO₂ uptake and release, as well as the size or functional differences in the carbon pools between the two scenarios. Therefore, we will keep both panels, but annotate both sides as per your request.

Specific comments:

Line 144: "August 2013 and June 2014, the year immediately following Blob onset, and recovered to pre-Blob concentrations by August 2014." Even though this is a citation, the comparison between June 2013 and 2014 would be more interesting to understand seasonality vs "blob" effects

Response: we have updated the sentence so it now reads: Total Chl *a* was approximately 50% lower in June 2014 than it was in June 2013, prior to the onset of the Blob, but had recovered to pre-Blob concentrations by August 2014¹⁴.

Line 168: This approach resolved 134 indicator OTUs associated with pre-Blob and Blob conditions (Fig. S4, S5, and Supplementary Table 1, 2) many of which would have been omitted solely on the basis of abundance filtering (Fig. S4). This sentence needs additional explanation. Why would these OTUs be excluded? What are these indicator OTUs indicative for?

Response: we agree that this sentence is confusing, thank you for pointing this out. We have re-phrased it to hopefully improve clarity. The section now reads: "Additionally, co-occurrence networks were constructed for the communities in both of these depth intervals to estimate connectivity among prokaryotic taxa in the 10-200 m zone at OSP. Following this, the results from the indicator analyses were filtered based on their selectivity (>3 positive edges), rather than a minimum relative abundance threshold. This approach allows for resolving potential key indicator OTUs without excluding conditionally rare taxa⁴², and resulted in 134 indicator OTUs associated with pre-Blob and Blob conditions (Fig. 5, S6, and supplementary Table 1, 2) which were selected for further analyses and discussion. "

Line 175: "Many of these taxa are known to directly interact with growing phytoplankton or sinking particulate organic matter and have been implicated in cofactor biosynthesis (e.g., B complex vitamins) or conversion of nitrogen (N) and sulfur (S) containing compounds produced during phytoplankton blooms." The problem with reporting results on the phylum level is that many microbial species have the mentioned traits both within these phyla and outside and vice versa many members of these phyla have completely different trades. Would it be possible to pinpoint at least specific genera to support these interpretations? Currently, it seems that possibly unrelated examples from these phyla are discussed.

Response: We completely agree with your statement regarding reporting results at the phylum level which is why, in the paragraphs following this introductory one, we unpack the indicators in more detail and discuss at class or order level, wherever possible.

Line 209: See line 175. This section suffers from the same problem as the one above.

Response: See comment above.

Line 280: The section "Food web structure and carbon export" would benefit strongly from an integration of the previously mentioned results. Which specific new results were used to design the conceptual model? Which parts of the model can be quantified? Which ones cannot right now?

Response: Thank you for pointing your concerns here. We have added text to help clarify. Most of this is described in detail in the supplemental material already, but the main text is indeed improved by a more detailed description of how we generated the conceptual model. The new text reads: "Using published values (please see chapter on model in the Supplementary Material for full details and reference list of all used published data), and the results of our amplicon sequencing of the prokaryotic community, we constructed a conceptual model of carbon flow in the OSP photic zone planktonic food web (Fig. 6). Where possible, we employed literature values to constrain carbon pools, but this was not always possible. For example, the viral component was estimated based on published virus to prokaryote ratios because no values of viral abundance exist for this region that we are aware of. Furthermore, measurements of growth or production, grazing and carbon

ECOSCOPE
routing the microcosmos

degradation are very rare in the NESAP region and thus we did not attempt to constrain the magnitude of connections between pools, other than qualitatively estimate whether they would be likely to increase or decrease under marine heatwave conditions. These data gaps highlight areas for future research, some of which currently are being addressed by the NASA Export Processes in the Ocean from Remote Sensing (EXPORTS) program⁹²⁻⁹⁴.”

REVIEWERS' COMMENTS:

Reviewer #2 (Remarks to the Author):

Traving et al. present a revised version of their manuscript "Prokaryotic responses to a warm temperature anomaly in northeast subarctic Pacific waters". I think the manuscript has improved a lot from its previous version and I only have minor comments.

1. As I mentioned previously it would be good if the manuscript would be quantitative, hence it would be good to add actual abundances and abundance shifts (at least for important taxa). I realize that the authors want to use information alternative to raw abundance for many analyses (such as the selection of indicator taxa) and I agree that this is a useful strategy. Nevertheless, the abundances are also important to understand the ecology and blob / non-blob shifts.
2. The figures have improved a lot, but I remain a bit critical of figure 4. I don't think that averaging over these heterogenic samples is too meaningful. Depth (as shown in the lower panel), but also seasonality and blob conditions are important parameters. Maybe subdividing the boxplots by season and/or blob could be beneficial.
3. In the section "Mixed responses and niche-differentiation", the authors mention that "none of the SAR202 indicators could be resolved below Order level." Even though this might be asking for a bit too much, I believe that a phylogenetic tree could be build from these OTUs to see if the response to the blob was mono- or polyphyletic.

Additional minor comments

In the description of figure 2, dotted lines are referred to, but I could not find them.